# Relationship between Psychological Needs and Academic Self-Concept in Physical Education Pre-Service Teachers: A Mediation Analysis

Antonio Granero-Gallegos [1,2], Ginés D. López-García [1,*], Antonio Baena-Extremera [3] and Raúl Baños [4,5,*]

1 Department of Education, University of Almeria, 04120 Almeria, Spain
2 Health Research Centre, University of Almeria, 04120 Almeria, Spain
3 Department of Musical, Plastic and Corporal Expression, Faculty of Education Sciences, University of Granada, 18071 Granada, Spain
4 Department of Musical, Plastic and Corporal Expression, Faculty of Social and Human Sciences, University of Zaragoza, Campus de Teruel, 44003 Zaragoza, Spain
5 Faculty of Sport, Autonomous University of Baja California, Tijuana 22390, Mexico
* Correspondence: glg807@ual.es (G.D.L.-G.); banos@unizar.es (R.B.)

**Abstract:** Academic self-concept plays a determining role in the teacher education process. Although research in education has focused on understanding the mechanisms that produce higher academic effort and academic self-confidence, the role that satisfaction and frustration of basic psychological needs and social support and resilience might play on academic self-concept is not known. The aim of the present study was to analyse the mediating effect of social support and resilience in the relationship between satisfaction/frustration of basic psychological needs and academic confidence and academic effort. A non-experimental, cross-sectional, correlational-causal study was designed. In total, 328 Physical Education pre-service teachers (131 women; 197 men) participated from eight Andalusian public universities. The age ranged from 22 to 45 years ($M$ = 24.83; $SD$ = 3.57). The following scales were used: Basic Psychological Needs Satisfaction in Education, Basic Psychological Needs Frustration in Education, Resilience, Social Support, and Academic Self-concept. A structural equations analysis with latent variables was carried out and the results obtained show that the satisfaction of basic psychological needs predicts an improvement in academic confidence and academic effort. In addition, resilience and social support significantly mediated the relationship between satisfaction of basic psychological needs and academic self-concept. This research also highlights the importance, both for teachers and researchers, of creating a context for encouraging the satisfaction of basic psychological needs, to promote academic self-concept in initial teacher training.

**Keywords:** academic confidence; academic effort; basic psychological needs; Self-determination Theory; physical education pre-service teachers

## 1. Introduction

Education must go beyond the teaching of theoretical content, acquiring skills, and evaluating the knowledge acquired [1] because, if we focus solely on these aspects, we would be treating students as if they were pieces on an assembly line [2,3]. Therefore, in the field of initial university training, qualities, such as having confidence in oneself and one's abilities, resilience, and making the effort to be a better professional, should be developed [4,5]. These qualities should be acquired by physical education teachers in their initial training (i.e., physical education pre-service teachers) as they play an important role in the character development and education of those children who will be part of a country's future [6]. In this way, future physical education teachers would be provided with tools that allow them to observe and reflect on educational situations in a professional manner and acquire the necessary basic skills during their initial training [7–9].

Among these capacities and skills to develop in initial training, academic self-concept (ASC) can be highlighted; this is understood as the perceptions a person has regarding their own skills in the academic field [10]. ASC is considered one of the most important indicators of educational quality, since the perceptions that students have regarding their academic abilities help to internalise those abilities and condition certain attitudes [11]. In this regard, the nature of the educational system itself is based on the premise of social comparison between students, especially in terms of the marks obtained [12]. This system is endorsed both by teachers and by the students' own families who, in many cases, pressure them into achieving success in school [13]. These comparisons between students can improve the ASC of those who achieve better academic results to the detriment of those who obtain worse scores [14] because, generally, students are forced to position themselves in relation to their peers through forced comparisons [13].

The ASC has been conceptualized as a two-dimensional construct comprising academic confidence and academic effort [15], and it has been proven that these two factors increase satisfaction and willingness to study among adolescent [11,16] and university students [17,18]. Academic confidence refers to the student's belief in the learning task and in achieving the learning goal; developing this confidence motivates students to carry out the learning tasks and achieve their goals [19]. However, university students with low confidence may make limited learning progress, experience a decline in academic performance, or even drop out [20]. Academic effort, on the other hand, is understood as the strength of mind and body that students must constantly apply during the academic year [21]; it is considered an important factor in the processes of adapting to university studies, academic performance, and self-efficacy in gaining employment after the degree course [22]. In contrast, students with low academic effort also present low expectations, confidence, and academic performance [23]. Research on academic confidence and academic effort in physical education pre-service teachers is scarce. Nonetheless, these are important skills to develop in teaching students as they will become the future trainers in a country's education system and, if they acquire these skills, they will be able to transmit them to their future students [24].

Recently, the importance of satisfaction of basic psychological needs (BPN) to improve ASC in primary school students has been demonstrated [13], although not with the aforementioned ASC factors (i.e., academic confidence and academic effort). These authors found that when students feel autonomous, competent, and connected to their teacher, it increases their confidence in carrying out learning activities, in their own potential for academic success, and in feeling a strong sense of personal value. Moreover, combining support practices for BPN satisfaction (SBPN) with meaningful, collaborative activities should also help to promote student learning proactivity and positive peer relationships [25,26]. BPNs are a theoretical construct of Self-determination Theory (SDT; ref. [27]). According to this theory, to achieve optimal functioning in school, the SBPN of the student's autonomy, competence and relatedness is necessary [27]. Conversely, the frustration of BPNs (FBPN) is linked to exhaustion, demotivation, and low academic engagement in university students [28,29]. It is worth mentioning that recent research has proposed novelty as another BPN [30], understanding this as both a unique memorable experience and as an everyday activity that serves to promote adaptive results [30]. However, the few studies that have linked BPN to ASC (e.g., [13,31]) did not include the novelty dimension. The present research does include it and proposes analysing the relationship between SBPN or FBPN and academic confidence and academic effort.

Several studies have found that resilience and the social support received are factors closely related to the SBPN or FBPN of university students [32–35]. Resilience, understood as the ability to absorb stressful situations, is one of the strategies used to improve the well-being of university students [35]. Promoting a culture of resilience in the learning environment and programmes aimed at fostering well-being and student care were initiatives carried out with medical students to great effect [36,37]. In this respect, Neufeld et al. [35] found that medical students who are resilient perceive SBPN, whereas those who have

not acquired a culture of resilience feel FBPN. The important role of resilience during the COVID-19 pandemic should also be highlighted, as it has been a factor for avoiding anxiety in stressful confinement situations [38–41]. However, few studies have analysed the relationship between BPN and resilience in physical education pre-service teachers or examined the mediating effect of resilience between BPN and academic confidence and academic effort. Therefore, we consider it relevant to study these relationships, given that physical education pre-service teachers need to develop high levels of confidence and effort to become the trainers of future professionals across many fields.

In addition, the role of the social support perceived by students during their academic life has also been shown to be important for achieving academic success [34]. Social support refers to the care and support that people feel from others [42], another variable that is closely related to BPN [34,43,44]; it has even been recognized as an important aspect of teacher resilience [45]. Studies have shown that when physical education pre-service teachers perceive social support from the family and/or their peers, they feel SBPN [34]. In this regard, various works have highlighted the importance of social support in students perceiving themselves as competent, and as an indicator of emotional health, since this increases occupational commitment, develops resistance in facing stressful situations, and encourages a positive attitude towards life [46]. Recently, it has been shown that when students perceive social support, their ASC increases [31], as does their academic confidence [47], and they make more effort academically when they feel support from the family [48]. Moreover, to the best of our knowledge, the link between social support and academic effort has only been studied in primary school students [48], and the research only analysed the perceived social support from family and peers [34,47,48], rather than perceived support from teachers. The present study analyses social support in terms of the support received from teachers (teachers' social support; see [49]). Therefore, we believe it is necessary to analyse the mediating effect of social support between the satisfaction/frustration of BPN and academic confidence and academic effort in pre-service teachers.

As one can see from the scientific literature mentioned above, there are few studies that analyse the possible predictive variables of effort and confidence in the training of physical education pre-service teachers. Prior research has analysed the relationship between SBPN and ASC [13], although in primary school students and without analysing the two main factors of ASC (i.e., academic effort and academic confidence), nor did it analyse how FBPN affects these variables. SBPN has also been linked to social support and resilience [33,34] without considering FBPN or the mediating effect of resilience and teacher social support between the satisfaction/frustration of BPN and academic effort and confidence among physical education pre-service teachers. In contrast, the present study does address these aspects. Therefore, we believe that our research provides evidence that fills a gap in the scientific literature and thus represents an interesting contribution. Consequently, the objective of this paper is to analyse the mediating effect of social support and resilience between the satisfaction/frustration of BPN and academic confidence/academic effort. A hypothesized model was created (see Figure 1) considering the postulates of the different theoretical currents. The following hypotheses were established: First, SBPN predicts academic confidence and academic effort (H1); second, FBPN negatively predicts academic confidence and academic effort (H2); third, social support positively mediates the relationship between SBPN and academic self-concept (H3); fourth, resilience positively mediates the relationship between SBPN and academic self-concept (H4); fifth, social support negatively mediates the relationship between SBPN and academic self-concept (H5); sixth, resilience negatively mediates the relationship between SBPN and academic self-concept (H6) (Figure 1).

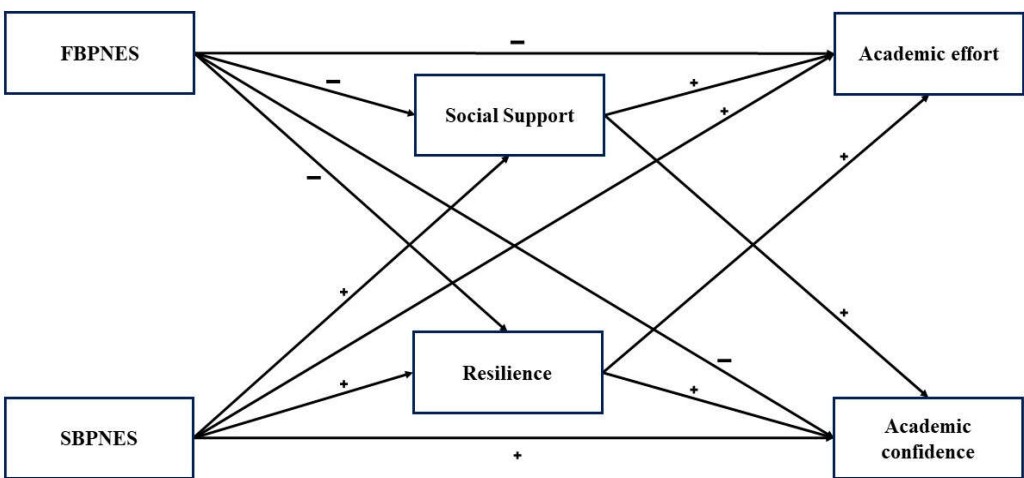

**Figure 1.** Hypothetical model with the expected relationships. Note: FBPN = Basic psychological needs frustration; SBPN = Basic psychological needs satisfaction.

## 2. Materials and Methods

### 2.1. Design

The research design was observational, descriptive, cross-sectional, and non-randomized. Students from eight Andalusian public universities participated. The following inclusion criterion was established: (i) to be a student of the Master's in Secondary Education Teaching in Physical Education at an Andalusian university. The exclusion criteria were: (i) failure to consent to the use of data in the study; (ii) not completing the data collection form.

### 2.2. Instruments

#### 2.2.1. Basic Psychological Needs Satisfaction in Education (SBPN)

We used the Spanish version adapted to the academic context [50] of the original scale by Gillet et al. [51]. The scale is composed of 15 items that are distributed over three dimensions of five items each: autonomy (e.g., "I feel free in giving my opinions"), competence (e.g., "I think I can respond to the demands of the subject programmes"), and relatedness to others (e.g., "I feel comfortable around others"). The five novelty satisfaction items (e.g., "I frequently feel there are novelties for me") [30] were integrated into this scale, as suggested by the authors themselves. A Likert scale between 1 (strongly disagree) and 5 (strongly agree) was used to collect the responses. SBPN was calculated as the mean value of the average scores of the factors that compose it.

#### 2.2.2. Basic Psychological Needs Frustration in Education (FBPN)

The Spanish version adapted to education [52] of the Psychological Need Thwarting Scale [53] was used. The scale comprises 12 items that measure the needs frustration of autonomy (four items, e.g., "I feel pushed to behave in certain ways"), competence (four items, e.g., "There are situations where I am made to feel incompetent"), and relatedness (four items, e.g., "I feel that other people dislike me"). The five items of novelty frustration (e.g., "I feel I do novel things") [30] were integrated into this scale, as proposed by the authors. A Likert scale between 1 (strongly disagree) and 5 (strongly agree) was used to collect the responses. FBPN was calculated as the mean value of the average scores of the factors that compose it.

#### 2.2.3. Resilience

The reduced version by Notario-Pacheco et al. [54] was used after being adapted to the Spanish university context. This scale comprises 10 items (e.g., "I can adapt to changes") that are organised into a dimension that measures resilience in young adults. A Likert scale

between 1 (never) and 5 (always) was used to collect the responses. High scores on the scale indicate a high level of resilience.

### 2.2.4. Social Support

The students' perceptions concerning the social support of their teachers were measured using three items (e.g., "I frequently feel there are novelties for me"); these were originally presented in the Social Support Questionnaire-6 [55], modified for use in education by Milton et al. [49], and used in the Spanish university context by Granero-Gallegos et al. [56]. A Likert scale between 1 (strongly disagree) and 5 (strongly agree) was used to collect the responses.

### 2.2.5. Academic Self-Concept

We used the Spanish version [17] of the Matovu scale [57]. The scale is composed of six items that measure academic confidence (three items, e.g., "I can follow the classes easily) and academic effort (three items, e.g., "I study hard for my tests"). A Likert scale between 1 (strongly disagree) and 7 (strongly agree) was used to collect the responses.

### *2.3. Procedure*

The academic heads of the Master's Degree in Secondary Education Teaching at the different Andalusian universities were contacted to inform them of the study objectives and to ask for their collaboration. Authorisation was obtained from the eight universities and a questionnaire was administered online during May 2021 and May 2022. In it, we explained the importance of the research, the anonymity of the responses, the way to fill in the scale, that participation in the study would not affect the students' qualification in any way, and that they could stop participating at any time. All participants gave prior consent for their responses to be included in the study. The research protocol was approved by the Bioethics Committee of the University of Almería (Ref:UALBIO2021/009).

### *2.4. Risk of Bias*

Regarding the risk of bias, it should be noted that there was blinding between the participants and the researchers who performed the data treatment and analysis. Regarding selection bias, participation in the study was voluntary and communication with students was carried out by email. There was no sample randomisation.

### *2.5. Sample Size*

A structural equations model (SEM) with six latent variables and 27 observable variables was conducted using the Free Statistics Calculator v.4.0 software [58], along with an a priori analysis of the sample size necessary to meet the study objective. It was calculated that a minimum of 320 students was needed to detect the effect sizes $f^2 = 0.245$, with a statistical power level of 0.90, and a significance level of $\alpha = 0.05$. A total of 328 students participated in the research.

### *2.6. Statistical Analysis*

SEM was performed to analyse how perceptions of basic psychological needs were associated with physical education pre-service teachers' resilience, social support, and academic self-concept. Following Wang et al. [59], the two-step method was used. The model fit was calculated, based on the values for the chi-square and degrees of freedom ($\chi^2/df$), the Comparative Fit Index (CFI), Tucker–Lewis Index (TLI), Root Mean Squared Error of Approximation (RMSEA) with its 90% confidence interval (CI), and the Standardised Root Mean Squared Residual (SRMR). For the $\chi^2/df$ ratio, values < 5.0 are considered acceptable, as are values > 0.90 for CFI and TLI, and values < 0.08 for RMSEA and SRMR [60,61]. Furthermore, values for the CFI of 0.90–0.94 and for the SRMR of less than 0.06 indicate an excellent fit [62]. Given the lack of multivariate normality (Mardia's coefficient = 15.57; $p < 0.001$), the maximum likelihood method was used with the bootstrapping procedure for

5000 re-samplings [63]. The reliability of each scale was evaluated using different parameters: composite reliability (CR), $\omega$ of McDonald, and AVE (Average Variance Extracted), to measure the convergent validity. Reliability values > 0.70 and AVE values > 0.50 are considered acceptable; however, according to Hair et al. [64], if all the factorial regression weights are significant and greater than 0.50, then we can assume that the factors have good convergent validity. In the present study, the regression weights for social support were between 0.57 and 0.76.

## 3. Results

### 3.1. Participants

A total of 328 university students of Physical Education participated (131 women; 197 men) from the Master's Degree in Secondary and Upper-Secondary Education Teaching, Vocational Training, and Language Teaching from eight Andalusian public universities (Spain) (University of Almeria, 20.4%, University of Cadiz, 8.8%, University of Cordoba, 2.4%, University of Granada, 34.1%, University of Huelva, 3%, University of Jaen, 1.5%, University of Malaga, 11.6%, and the University of Seville, 18%). The age of the participants was between 22 and 45 years ($M = 24.83$; $SD = 3.57$). There were no missing values in the included sample data.

### 3.2. Preliminary Analysis

The descriptive statistics and the correlations between the different variables are shown in Table 1.

**Table 1.** Descriptive statistics and correlations between variables.

| Variable | Range | M | SD | Q1 | Q2 | ω | CR | AVE | 2 | 3 | 4 | 5 | 6 |
|---|---|---|---|---|---|---|---|---|---|---|---|---|---|
| FBPN | 1–5 | 2.16 | 0.78 | 0.65 | −0.01 | 0.85 | 0.91 | 0.73 | −0.58 ** | −0.38 ** | −0.34 ** | −0.27 ** | −0.48 ** |
| SBPN | 1–5 | 3.88 | 0.65 | −0.22 | 0.10 | 0.84 | 0.86 | 0.61 | - | 0.53 ** | 0.46 ** | 0.45 ** | 0.51 ** |
| Resilience | 1–5 | 4.24 | 0.60 | −0.61 | 0.41 | 0.88 | 0.88 | 0.54 | | - | 0.29 ** | 0.32 ** | 0.46 ** |
| Social Support | 1–5 | 3.62 | 0.86 | −0.24 | −0.42 | 0.71 | 0.71 | 0.47 | | | - | 0.28 ** | 0.37 ** |
| Academic Effort | 1–7 | 5.36 | 1.29 | −0.90 | 0.54 | 0.81 | 0.73 | 0.51 | | | | - | 0.35 ** |
| Academic Confidence | 1–7 | 5.96 | 0.96 | −1.43 | 1.25 | 0.72 | 0.82 | 0.61 | | | | | - |

Note. ** The correlation is significant at the 0.01 level; $M$ = Mean; $SD$ = Standard Deviation; SBPN = Satisfaction of Basic Psychological Needs; FBPN = Frustration of Basic Psychological Needs; Q1 = Skewness; Q2 = Kurtosis; $\omega$ = Omega of McDonald; CR = Composite Reliability; AVE = Average Variance Extracted.

### 3.3. Main Analysis

In step 1, the SEM presented excellent goodness of fit indices: $\chi^2/\mathrm{df} = 2.07$, $p = 0.066$; CFI = 0.99; TLI = 0.97; RMSEA = 0.057 (90%CI = 0.000; 0.107; $p_{close} = 0.342$), SRMR = 0.031. In step 2, the hypothesised SEM presented a similar excellent fit: $\chi^2/\mathrm{df} = 2.07$, $p = 0.066$; CFI = 0.99; TLI = 0.97; RMSEA = 0.057 (90%CI = 0.000; 0.107; $p_{close} = 0.342$); SRMR = 0.031.

The relationships between the satisfaction/frustration of the BPNs and the two academic self-concept factors (i.e., confidence and effort), as well as the relationships between the BPN and the mediators (i.e., resilience and social support), and between the mediators and academic confidence and academic effort, can be seen in Figure 2 and Table 2.

The model achieved an explained variance of 36% for confidence, 21% for effort, 28% for resilience, and 21% for social support (Figure 2). The model shows that the FBPN does not have a statistically significant positive predictive relationship with any of the variables studied; however, the direct effect on academic confidence was negative and statistically significant ($p = 0.019$). In contrast, the SBPN presents a direct, positive predictive relationship with social support ($p = 0.005$), resilience ($p = 0.014$), effort ($p = 0.012$), and confidence ($p = 0.040$). Likewise, resilience has a positive direct effect on confidence ($p = 0.008$), but has no direct relationship with effort. In contrast, social support directly predicts both effort ($p = 0.024$) and confidence ($p = 0.009$). Regarding mediation, resilience

and social support act as a positive mediating variable between SBPN and confidence (*p* = 0.007); although, the most important effect of these variables is that they increase the total effect of SBPN on confidence (*p* = 0.004). On the other hand, social support acts as a mediator between SBPN and effort (*p* = 0.007), though, in this case, it also plays a relevant role in increasing the total effect of SPBN on effort (*p* = 0.021). In addition, Figure 2 shows the CI (95%) of $R^2$, confirming that they can be taken as measures of ES (Domínguez-Lara, 2017) [65], and it should be noted that the ES in all the analysed relationships is large [66].

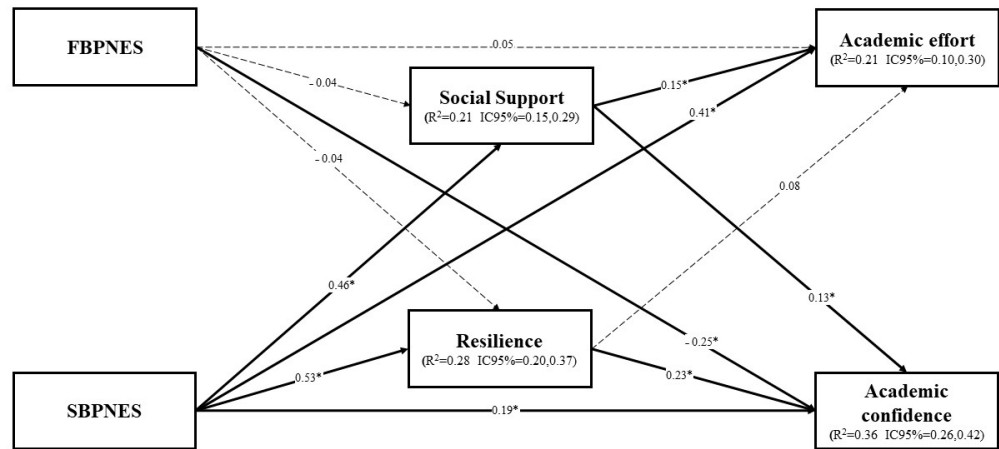

**Figure 2.** Predictive relationships of the frustration and satisfaction of basic psychological needs with academic confidence and academic effort through the mediating role of resilience and social support. Note: * *p* < 0.05. FBPN = Basic psychological needs frustration; SBPN = Basic psychological needs satisfaction. $R^2$ = Explained variance; CI = Confidence interval. The dashed lines represent non-significant relationships.

**Table 2.** Estimation of significant standardised parameters and statistics of the mediation model.

| Independent Variable | Dependent Variable | Mediator | β | SE | 95% CI | |
|---|---|---|---|---|---|---|
| | | | | | Inf | Sup |
| **Direct Effects** | | | | | | |
| FBPN | Academic confidence | | −0.25 * | 0.07 | −0.36 | −0.10 |
| SBPN | Resilience | | 0.53 * | 0.04 | 0.45 | 0.61 |
| SBPN | Academic confidence | | 0.19 * | 0.09 | 0.02 | 0.39 |
| SBPN | Social support | | 0.46 * | 0.06 | 0.39 | 0.54 |
| SBPN | Academic effort | | 0.41 * | 0.09 | 0.29 | 0.51 |
| Resilience | Academic confidence | | 0.23 * | 0.08 | 0.11 | 0.36 |
| Social support | Academic confidence | | 0.13 * | 0.06 | 0.04 | 0.24 |
| Social support | Academic effort | | 0.15 * | 0.08 | 0.02 | 0.27 |
| Indirect effects | | | | | | |
| SBPN | Academic confidence | Resilience | 0.13 * | 0.06 | 0.10 | 0.27 |
| SBPN | Academic confidence | Social support | 0.06 * | 0.05 | 0.02 | 0.23 |
| SBPN | Academic effort | Social support | 0.07 * | 0.06 | 0.03 | 0.25 |
| Total Effects | | | | | | |
| SBPN | Academic confidence | | 0.37 * | 0.06 | 0.24 | 0.54 |
| SBPN | Academic effort | | 0.45 * | 0.05 | 0.31 | 0.54 |

Note. β = Estimation of standardised parameters; SE = standard error; 95%CI = 95% confidence interval; Inf = Lower limit of 95% CI; Sup = Upper limit of 95% CI; * *p* < 0.05.

## 4. Discussion

The objective of this research was to analyse the mediating effect of social support and resilience between satisfaction/frustration of BPN and academic confidence/academic effort in physical education pre-service teachers. The main results show that SBPN has a

direct and positive effect on confidence and effort. The total effects on these two variables are increased with the mediation of social support, while the mediation of resilience only increases the total effect on confidence. FBPN is not significantly related to any of the variables studied.

To the best of our knowledge, no research has previously linked BPN (satisfaction/frustration) to academic confidence/academic effort in university students. The present work shows that SBPN directly and positively predicts confidence in physical education pre-service teachers. These results follow the trend of other research studies in which SBPN improved variables, such as self-confidence in university students [67], self-esteem in secondary school students, and self-concept in primary school students [13]. Thus, as indicated by Guay et al. [25], practices supporting SBPN that include meaningful and collaborative activities should be implemented in the classroom to promote proactivity towards learning and positive relationships with peers. In addition, this will augment confidence and a healthy motivational orientation that results in student well-being, enjoyment, persistence, and engagement [27]. Since the BPN theory is a construct that shapes motivation [27], it is worth mentioning that there are significant mutual influences between an individual's learning achievements, motivation, and confidence [68]. On the other hand, those students who do not have sufficient academic confidence lose belief in their ability to solve tasks [69] and their interest in learning, increasing the possibility that they will abandon their studies [70]. Therefore, it is important to develop academic confidence, as it helps one to acquire learning strategies and skills effectively [71].

The results of this study also show that the total effects of SBPN on academic confidence are increased by mediation, both by the perceived support of the teacher (i.e., social support) and by resilience. We are not aware of any prior research that has linked these variables. In the case of teacher support, we can say that our results align with the study by Park and Hong [47], who analysed the effect of family social support on university nursing students, indicating that this support positively predicted academic confidence. In other contexts, teacher social support has been positively linked to SBPN and academic satisfaction, and to ASC in secondary school students [31]. Social support from family and friends has also been linked to improved ASC in first-year university students [72]. These results may be because, when students feel supported by their teachers and perceive the teachers showing concern for their learning, this can increase their expectation of success and, according to Chemers et al. [73], this expectation of success is closely linked to academic confidence. These factors are important in physical education pre-service teachers because they will form part of the future educational system of the country. Furthermore, the application of knowledge depends on skills and competence and, in turn, applying these requires confidence.

Regarding the mediating role of resilience, several studies have shown that SBPN positively predicts resilience [32–35], and that resilience predicts confidence in university nursing students [74]. This relationship might be because, when teachers promote SBPN in students and educate them to manage the stressful situations they may face as future teachers, they are providing them with psychological and affective skills that increase their academic confidence. Therefore, it is important to promote a resilient culture within the learning environment, along with programmes in which university students learn to deal with stressful situations [37]. This is important because students who lose confidence in their ability to succeed academically are more likely to fail, even if they have good academic skills.

One of the striking results of this study is that resilience, which does predict confidence, is not a predictor of academic effort. On the other hand, both SBPN and social support show a predictive and positive relationship with effort. In this case, social support mediation also increases the overall effects of SPBN on effort. No studies have been found that analyse the relationship of the aforementioned variables; however, motivation is a construct closely linked to BPN [27], and it has been shown that teachers have to previously motivate students to increase their academic effort [75]. In addition, effort has been related to better

motivational regulation in secondary school students [76], with a higher level of interest and learning awareness in university students [75], with greater use of ICTs in educational innovation [77], and with increased chances of success in finding work following the degree [22]. It would appear to be important that future teachers experience support from their own teachers during initial training so that, later in their professional career, they can offer support to all their own students, this being an engine of global change.

Regarding the direct effects of FBPN on confidence and effort, or the indirect effects through resilience and social support, no statistically significant relationships were found in this research. There are few studies that relate these variables and, moreover, our research results do not follow the same line as previous works. Several studies negatively associated FBPN with resilience [35] and with social support [78], although it should be noted that these studies analysed the social support received from family and friends, not from teachers. FBPN has also been linked to worsening self-esteem [79], with a lesser passion for study that negatively influences study strategies [80], and with less positive psychological adjustment in the first year of university [32]. The difference in results between this study and the existing scientific literature may be due to the confinement and restrictions caused by the COVID-19 pandemic [81]. Given that the data were collected a few months after home confinement, but in the middle of the pandemic, it is possible that, when returning to class in person, students were satisfied with their BPN (i.e., SBPN), due to the desire to return to "normality", and that less BPN frustration was perceived (i.e., FBPN). Because of this question and the scarce scientific literature, we believe that caution should be exercised when interpreting the results of this study and that further in-depth research should be pursued along this line.

### 4.1. Limitations and Future Research

As its strengths, we can highlight the topic investigated, the sample size used (one third of the total population), the level of confidence (95%) and the margin of error (2.23%). The study contributes to the scientific literature in the field of physical education pre-service teachers because, as far as we are aware, the relationships between some of these variables (e.g., academic confidence and academic effort) have not been previously studied. Another strength is the type of analysis undertaken, since an SEM has been carried out with latent variables; however, the direction of causality cannot be determined, and reverse causality is a possibility as cross-sectional studies cannot establish a causal relationship between the predictor and outcome variables, making it difficult to determine the direction of causality in a mediational relationship. At the same time, the study's strength can become other limitations. As we have indicated throughout, the scarcity of research relating the analysed variables to confidence and effort makes it necessary to interpret the results with caution. As well, the present research includes a cross-sectional design and the results and measurements were obtained via questionnaires. Due to the above limitations, we consider it necessary for future work to look at this line of research in more detail to see if the study results are supported, establishing longitudinal and/or experimental designs, and adding observational instruments that measure the influence of BPN on academic confidence and academic effort in teacher training.

### 4.2. Practical Implications

These research results underline the importance of SBPN, resilience, and the social support provided by university professors to physical education pre-service teachers in order for them to have academic confidence and academic effort [82]. Therefore, it is recommended that those professors in charge of training future teachers receive instruction on how to teach educational strategies that promote novel and innovative learning environments, achieve learning goals, and enhance communication and cooperation between students so that they can make decisions autonomously and feel competent in this context [83,84]. University professors not only have to create these learning environments, but also make physical education pre-service teachers aware of the learning tools being used,

how they can be applied with their future students, and provide individualised teaching that is adapted to the needs of the students. In addition, as proposed by authors, such as Neufeld et al. [35], students should be educated in resilience, that is, how to handle stressful situations (e.g., to remain calm when conflicts arises in the classroom, seeking different ways of resolving them) and, as proposed by authors, such as Schwabe et al. [31], physical education pre-service teachers should be taught to support their future students (e.g., to search for new learning strategies when a group of students is failing to achieve the learning goal).

## 5. Conclusions

In summary, SBPN has been shown to improve academic confidence and academic effort in physical education pre-service teachers. Furthermore, confidence is increased when students are resilient and when they feel supported by the teacher (i.e., social support) during the sessions; this teacher support is also important for students to try harder in class. In this sense, the effort, as well as the academic confidence of pre-service PE teachers, will be stimulated when in a teacher education context, when the psychological needs of the students feel satisfied, and also when the pre-service teachers perceive social support from the teacher educator and show a resilient attitude in the development of the sessions. On the contrary, the perception of frustration in learners' psychological needs will reduce their confidence. Finally, teacher trainers should incorporate methodological strategies in class based on novel tasks that develop their students' perception of competence, increase their feeling of autonomy in making decisions, promote affective bonds between them, and focus on increasing their ability to deal with stressful situations, making them aware that they have the support of the teachers.

**Author Contributions:** Conceptualisation, G.D.L.-G., A.G.-G. and R.B.; methodology, A.G.-G. and A.B.-E.; formal analysis, A.G.-G. and A.B.-E.; investigation, G.D.L.-G., A.G.-G., R.B. and A.B.-E.; data curation, A.G.-G. and R.B.; writing—original draft preparation, G.D.L.-G., A.G.-G., A.B.-E. and R.B.; writing—review and editing, G.D.L.-G., A.G.-G., A.B.-E. and R.B.; project administration, A.G.-G. All authors have read and agreed to the published version of the manuscript.

**Funding:** This work was carried out thanks to the help received from the "I + D + i" research project entitled: "Is the empowering-disempowering motivational climate that undergraduate students perceive related to their intention to become teachers? A longitudinal study with teachers in training" (Ref. P20_00148), funded by the Andalusian Plan for Research, Development, and Innovation (PAIDI, 2020) of the Junta de Andalucía and help to research projects from the Health Research Center of the University of Almería. As well, this article was carried out during a research stay of Dr. Antonio Granero-Gallegos at the University of Granada (from 14 October 2022, to 14 January 2023) with Dr. Antonio Baena-Extremera.

**Institutional Review Board Statement:** The study was conducted in accordance with the Declaration of Helsinki and approved by the Bioethics Committee of UNIVERSITY OF ALMERIA (protocol code UALBIO2021/009, 17 February 2022).

**Informed Consent Statement:** Informed consent was obtained from all subjects involved in the study.

**Data Availability Statement:** The data presented in this study are available on request from the corresponding author. The data are not publicly available due to privacy.

**Conflicts of Interest:** The authors declare no conflict of interest.

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
