# Peer review of "Relationship between Psychological Needs and Academic Self-Concept in Physical Education Pre-Service Teachers: A Mediation Analysis"

_sustainability, doi:10.3390/su15054052_

Round 1

Reviewer 1 Report

Thank you for the opportunity to review this interesting article entitled: Relationship between psychological needs and academic self-concept in Physical Education pre-service teacher: A mediation analysis.

The manuscript is of great interest and is well structured and theoretically justified. The aim is pertinent, and the hypotheses are justified. Likewise, the methodology used to answer the objective is adequate and very well explained. The sample is very large, and the practical implications proposed can also be highlighted.

In this reading, the aim of this study was to analyse the mediating effect of social support and resilience in the relationship between satisfaction/frustration of basic psychological needs and academic confidence and academic effort. A non-experimental, cross-sectional, correlational-causal study was designed. In total, 328 Physical Education pre-service teachers (131 women; 197 men) participated from eight Andalusian public universities. The age ranged from 22 to 45 years (M=24.83; SD=3.57). The following scales were used: Basic Psychological Needs Satisfaction in Education, Basic Psychological Needs Frustration in Education, Resilience, Social Support, and Academic Self-concept. A structural equations analysis with latent variables was carried out and the results obtained show that the satisfaction of basic psychological needs predicts an improvement in academic confidence and academic effort. Confidence is increased when students are resilient and when they feel supported by the teacher; this support is also important for students to try harder in class.

I highlight and agree with the results of this study, as they reveal the influence of a satisfaction/frustration of basic psychological needs in the prediction of academic confidence and effort in pre-service teachers. Besides, on a formal level, it is very well structured, with clear and concise language. It uses current bibliographical references for the subject matter.

Congratulation for the work. However, I would ask you to correct the following:

-       In the title: teachers instead teacher?

-       The Hypothetical model (Figure 1) was developed by the author or adopted by other researchers. This needs additional explanation so that future researchers can easily follow it.

-       Revise “note” of the Figure 1. There is some acronyms that do not correspond with the Figure. Likewise, could you avoid acronyms in the Figures?

-       In the participants, please, specify the proportion of the sample from eight Andalusian public universities.

-       In the References section, those that are in Spanish, should be followed by their English translation in square brackets [English translation]. For example, reference: 65.

Author Response

We thank the reviewers for his/her constructive comments and his/her thorough revision of the manuscript. Below we answer his/her questions and concerns, including explicitly the changes made in the manuscript as well.

I highlight and agree with the results of this study, as they reveal the influence of satisfaction/frustration of basic psychological needs in the prediction of academic confidence and effort in pre-service teachers. Besides, on a formal level, it is very well structured, with clear and concise language. It uses current bibliographical references for the subject matter.

Congratulation on the work. However, I would ask you to correct the following:

Comment-1.  In the title: teachers instead teacher?

  • Response: Suggestions made by the reviewer were made.

Comment-2. The Hypothetical model (Figure 1) was developed by the author or adopted by other researchers. This needs additional explanation so that future researchers can easily follow it.

  • Response: Yes, the Hypothetical model (Figure1) was developed by the author, and the following hypotheses were established, so future researchers can follow.

“A hypothesized model was created (see Figure 1) taking into account the postulates of the different theoretical currents. The following hypotheses were established: First, SBPN predicts academic confidence and academic effort (H1); second, FBPN negatively predicts academic confidence and academic effort (H2); third, social support positively mediates the relationship between SBPN and academic self-concept (H3); fourth, resilience positively mediates the relationship between SBPN and academic self-concept (H4); fifth, social support negatively mediates the relationship between SBPN and academic self-concept (H5); sixth, resilience negatively mediates the relationship between SBPN and academic self-concept (H6) (Figure 1).”

Comment-3. Revise the “note” of Figure 1. There is some acronyms that do not correspond with the Figure. Likewise, could you avoid acronyms in the Figures?

  • Response: Suggestions made by the reviewer were made. Acronyms in the “note” of Figure 1 have been revised.

Comment-4. In the participants, please, specify the proportion of the sample from eight Andalusian public universities.

  • Response: Suggestions made by the reviewer were made and proportions of the sample from eight Andalusian public universities have been added in the paragraph: “participants”.

Comment-5. In the References section, those that are in Spanish should be followed by their English translation in square brackets [English translation]. For example, reference: 65.

  • Response: Suggestions made by the reviewer were made.

Reviewer 2 Report

Dear,

The article needs small ones that were pointed out in the text. The summary remains to be revised. It is also necessary that the conclusion be deepened on the subject. It remains to insert the limitations of the work and future works.

Author Response

Comments and Suggestions for Authors

The article needs small ones that were pointed out in the text. The summary remains to be revised. It is also necessary that the conclusion be deepened on the subject. It remains to insert the limitations of the work and future works.

Comment-1. It remains to insert the research problem and also the results of the work.

  • Response: Suggestions made by the reviewer were made, so the abstract presents a brief introduction to the research problem and the main results of them. “Academic self-concept plays a determining role in the teacher education process. Although research in education has focused on understanding the mechanisms that produce higher academic effort and academic self-confidence, the role that satisfaction and frustration of basic psychological needs and social support and resilience might play on academic self-concept is not known. The aim of the present study was to analyse the mediating effect of social support and resilience in the relationship between satisfaction/frustration of basic psychological needs and academic confidence and academic effort. A non-experimental, cross-sectional, correlational-causal study was designed. In total, 328 Physical Education pre-service teachers (131 women; 197 men) participated from eight Andalusian public universities. The age ranged from 22 to 45 years (M=24.83; SD=3.57). The following scales were used: Basic Psychological Needs Satisfaction in Education, Basic Psychological Needs Frustration in Education, Resilience, Social Support, and Academic Self-concept. A structural equations analysis with latent variables was carried out and the results obtained show that the satisfaction of basic psychological needs predicts an improvement in academic confidence and academic effort. In addition, resilience and social support significantly mediated the relationship between satisfaction of basic psychological needs and academic self-concept. This research also highlights the importance, both for teachers and researchers, of creating a context for encouraging the satisfaction of basic psychological needs, to promote academic self-concept in initial teacher training.

Comment-2. I missed a more in-depth conclusion of the theme. Missing to insert the limitations of the work and future work.

  • Response: First, the limitations and future perspectives of the research can be easily found in section 4.1. Furthermore, more in-depth conclusions of the work were written in section 5. “In summary, SBPN has been shown to improve academic confidence and academic effort in physical education pre-service teachers. Furthermore, confidence is increased when students are resilient and when they feel supported by the teacher (i.e., social support) during the sessions; this teacher support is also important for students to try harder in class. In this sense, the effort, as well as the academic confidence of pre-service PE teachers, will be stimulated when in a teacher education context, the psychological needs of the students feel satisfied and also when the pre-service teachers perceive social support from the teacher educator and show a resilient attitude in the development of the sessions. On the contrary, the perception of frustration in learners' psychological needs will reduce their confidence. Finally, teacher trainers should incorporate methodological strategies in class based on novel tasks that develop their students’ perception of competence, increase their feeling of autonomy in making decisions, promote affective bonds between them, and focus on increasing their ability to deal with stressful situations, making them aware that they have the support of the teachers.”

Reviewer 3 Report

Thank you so much for inviting me to review this manuscript. Authors tried to to analyse the mediating effect of social support and resilience between the satisfaction/frustration of BPN and academic confidence/academic effort. Mediation analysis with cross-sectional designs are often considered problematic because cross-sectional studies provide a snapshot of data at a single point in time, rather than capturing the change over time. This means that the direction of causality cannot be determined and reverse causality is a possibility. In other words, cross-sectional studies cannot establish a causal relationship between the predictor and outcome variables, making it difficult to determine the direction of causality in a mediational relationship. Additionally, cross-sectional studies do not allow for the examination of temporal relationships, which are often critical in mediational analysis. It is recommended to use a longitudinal design in order to establish causal relationships in mediational analysis. 

Best wishes,

Author Response

Comments and Suggestions for Authors

Thank you so much for inviting me to review this manuscript. Authors tried to to analyse the mediating effect of social support and resilience between the satisfaction/frustration of BPN and academic confidence/academic effort. Mediation analysis with cross-sectional designs are often considered problematic because cross-sectional studies provide a snapshot of data at a single point in time, rather than capturing the change over time. This means that the direction of causality cannot be determined and reverse causality is a possibility. In other words, cross-sectional studies cannot establish a causal relationship between the predictor and outcome variables, making it difficult to determine the direction of causality in a mediational relationship. Additionally, cross-sectional studies do not allow for the examination of temporal relationships, which are often critical in mediational analysis. It is recommended to use a longitudinal design in order to establish causal relationships in mediational analysis. 

  • Response: Thank you very much for your comments. Of course, we agree with this comment of the reviewer. The direction of causality cannot be determined and reverse causality is a possibility and cross-sectional studies cannot establish a causal relationship between the predictor and outcome variables, making it difficult to determine the direction of causality in a mediational relationship. This comment has been added in the paragraph “limitations and future research”.

However, in this paper, we refer to the prediction relationship between variables and the direction of the arrows based on the revised literature, as you can check in the hypothetical model (Figure 1). For this reason, we believe that this SEM can add more scientific literature on these variables and the direction of the causality.

Similar analyzes (SEM) with cross-sectional studies in Physical Education are usually published by different authors in prestigious journals. Let us give some recent examples:

Abós, A.; Burgueño, R.; García-González, L.; Sevil-Serrano, J. Influence of Internal and External Controlling Teaching Behaviors on Students’ Motivational Outcomes in Physical Education: Is There a Gender Difference? J. Teach. Phys. Educ. 2022, 41(3), 502-512. https://doi.org/10.1123/jtpe.2020-0316

Franco, E.; Cuevas, R.; Coteron, J.; Spray, C. Work Pressures Stemming From School Authorities and Burnout Among Physical Education Teachers: The Mediating Role of Psychological Needs Thwarting J. Teach. Phys. Educ. 2022, 41(1), 110-120. https://doi.org/10.1123/jtpe.2020-0070

García-González, L.; Sevil-Serrano, J.; Abós, A.; Aelterman, N.; Haerens, L. The role of task and ego-oriented climate in explaining students’ bright and dark motivational experiences in Physical Education. Phys. Educ. Sport Pedag. 2019, 24:4, 344-358. https://doi.org/10.1080/17408989.2019.1592145

Granero-Gallegos, A.; Hortigüela-Alcalá, D.; Hernández-Arijo, A.; Carrasco-Poyatos, M. Teaching style and competence in higher education: the motivational climate mediation. Educ. XX1 2021, 24(2), 43-64. https://doi.org/10.5944/educXX1.28172

Mastagli, M.; Van Hoye, A.; Hainaut, J.P.; Bolmont, B. The Role of an Empowering Motivational Climate on Pupils' Concentration and Distraction in Physical Education. J. Teach. Phys. Educ. 2022, 41(2), 311-321. https://doi.org/10.1123/jtpe.2020-0252

Sidala, H.; Koka, A. Gender Differences in the Relationships Between Perceived Teachers’ Controlling Behaviors and Amotivation in Physical Education. J. Teach. Phys. Educ. 2018, 37(2), 197-208. https://doi.org/10.1123/jtpe.2017-0199

Even, recent research (from several scientific fields) has been published in Sustainability with SEM:

Ghodsi, M.; Pourmadadkar, M.; Ardestani, A.; Ghadamgahi, S.; Yang, H. Understanding the Impact of COVID-19 Pandemic on Online Shopping and Travel Behaviour: A Structural Equation Modelling Approach. Sustainability 202214, 13474. https://doi.org/10.3390/su142013474

Ren, J.; Su, K.; Zhou, Y.; Hou, Y.; Wen, Y. Why Return? Birdwatching Tourists’ Revisit Intentions Based on Structural Equation Modelling. Sustainability 202214, 14632. https://doi.org/10.3390/su142114632

Persada, S.F.; Prasetyo, Y.T.; Suryananda, X.V.; Apriyansyah, B.; Ong, A.K.S.; Nadlifatin, R.; Setiyati, E.A.; Putra, R.A.K.; Purnomo, A.; Triangga, B.; Siregar, N.J.; Carolina, D.; Maulana, F.I.; Ardiansyahmiraja, B. How the Education Industries React to Synchronous and Asynchronous Learning in COVID-19: Multigroup Analysis Insights for Future Online Education. Sustainability 202214, 15288. https://doi.org/10.3390/su142215288

Round 2

Reviewer 3 Report

Thank you for your response. Just because something has been published many times does not make it valid. With this type of design, it is not appropriate to perform this type of study. Sustainability should be more rigorous with the type of articles that the authors indicate. They are all cross-sectional in design and have the same limitation. Another question, why did not the authors send this paper to a specific psychology journal. Even more so when they talk about them being theoretical models.

Here you have another reference justifying my position:

https://academic.oup.com/ajcn/article/105/6/1259/4569799

Best wishes,

Author Response

We again thank the reviewer for the response. To our previous response, we would like to add that the psychometric scientific literature, specialized in SEM (e.g., Byrne, 2012; Kline, 206; Hair et al., 2018) supports the use of this technique in cross-sectional studies, because it is a technique that It is based on testing direct and indirect predictive effects between a series of independent (i.e., predictive), moderator and dependent variables. Indeed, cause-effect relationships cannot be established, but since it is based on previous theory, it is this theory that gives coherence and theoretical rationality to the sense of the relationships of the hypothesized model that does contribute to the scientific literature.

  1. Kline, R. B. Principles and practice of structural equation modeling, 4th ed.; Guilford Press; New York, USA, 2016.
  2. Hair, J. F.; Black, W. C.; Babin, B. J.; Anderson, R. E. Multivariate Data Analysis, 8th ed.; Pearson. 2018.
  3. Byrne, B. M. Structural Equation Modeling with AMOS, 2nd. Ed.; Routledge. 2011.

Since we do not want to extend ourselves, because it is not a question of convincing, but of arguing, and we have already given several examples of the numerous literature published with SEM and cross-sectional design, we are only going to provide two more examples of publications.

1.- Published article in a specific sports psychology journal (Journal: PSYCHOLOGY OF SPORT AND EXERCISE; Impact Factor 2021: 5.118; JCR-Rank: 18/80; Q1; Category: Psychology). In addition, the article is also signed by one of the most prestigious authors worldwide in psychology (Self-Determination Theory), Edward L. Deci.

  • Behzadnia, B.; Adachi, P.J.C.; Deci, E.L.; Mohammadzadeh,H. Associations between students' perceptions of physical education teachers' interpersonal styles and students' wellness, knowledge, performance, and intentions to persist at physical activity: A self-determination theory approach. Psychology of Sport and Exercise, 2018, 39, 10-19. https://doi.org/10.1016/j.psychsport.2018.07.003

2.- Published article (i.e., Stupnisky et al., 2018) in one of the most prestigious journals in the category of educational psychology. Journal: CONTEMPORARY EDUCATIONAL PSYCHOLOGY; Impact Factor 2021: 6.922; JCR-Rank: 3/61 (Q1); Category: Psychology, Educational.

  • Stupnisky, R. H.; BrckaLorenz, A.; Yuhas, B.; Guay, F. Faculty members’ motivation for teaching and best practices: Testing a model based on self-determination theory across institution types. Contemporary Educational Psychology, 2018, 53, 15-26. https://doi.org/10.1016/j.cedpsych.2018.01.004
